# Modeling Yields Response to Shading in the Field-to-Forest Transition Zones in Heterogeneous Landscapes

**Martin Schmidt** [1,2,*] , **Claas Nendel** [1] , **Roger Funk** [1] , **Matthew G. E. Mitchell** [3] **and Gunnar Lischeid** [1,2]

1   Leibniz Centre for Agricultural Landscape Research, Eberswalder Straße 84, 15374 Müncheberg, Germany; nendel@zalf.de (C.N.); funk@zalf.de (R.F.); lischeid@zalf.de (G.L.)
2   Institute of Earth and Environmental Sciences, University of Potsdam, Karl-Liebknecht-Str. 24-25, 14476 Potsdam, Germany
3   Institute for Resources, Environment and Sustainability, University of British Columbia, 429-2202 Main Mall, Vancouver, BC V6T 1Z4, Canada; matthew.mitchell@ubc.ca
*   Correspondence: martin.schmidt@zalf.de; Tel.: +49-33432-82-379

**Abstract:** In crop modeling and yield predictions, the heterogeneity of agricultural landscapes is usually not accounted for. This heterogeneity often arises from landscape elements like forests, hedges, or single trees and shrubs that cast shadows. Shading from forested areas or shrubs has effects on transpiration, temperature, and soil moisture, all of which affect the crop yield in the adjacent arable land. Transitional gradients of solar irradiance can be described as a function of the distance to the zero line (edge), the cardinal direction, and the height of trees. The magnitude of yield reduction in transition zones is highly influenced by solar irradiance—a factor that is not yet implemented in crop growth models on a landscape level. We present a spatially explicit model for shading caused by forested areas, in agricultural landscapes. With increasing distance to forest, solar irradiance and yield increase. Our model predicts that the shading effect from the forested areas occurs up to 15 m from the forest edge, for the simulated wheat yields, and up to 30 m, for simulated maize. Moreover, we estimated the spatial extent of transition zones, to calculate the regional yield reduction caused by shading of the forest edges, which amounted to 5% to 8% in an exemplary region.

**Keywords:** edge effect; transition zone; solar irradiance; crop growth; maize; wheat

## 1. Introduction

Food provision is a fundamental ecosystem service with an emerging importance. Model-based projections of agricultural yields, as an indicator for food availability, have developed into an essential tool to derive strategies for a sustainable food supply, to meet the demands of an increasing world population. Crop models that are developed for this purpose often simulate single plants that represent a certain area. These models usually incorporate management activities, climate, and soil conditions. However, these represented areas are often not put into the landscape context, and it is not clear, until now, which important feedback mechanisms the models are not able to capture, for this reason. The heterogeneity of agricultural landscapes, in terms of land cover, as well as landscape structures and elements (e.g., forested areas, trees, and hedges), impose different effects on the plants that grow there. These landscape elements differ from arable land, in their physical or biological nature, as they host different species or receive different types of management. One obvious difference is the height of trees and shrubs, compared to smaller crops of arable land, a feature that dominates the visible appearance of such landscape elements. The height difference affects the physical conditions of the habitats, at the

edge between such landscape elements and the surrounding agricultural land, in which crops grow to a maximum of 2 to 3 m height. These transition zones [1] and their specific environment, are expected to play a vital role in the productivity and biodiversity of agricultural landscapes. Hence, crop models need to include these transition zones if they want to accurately model crop yields across space.

Forested areas, trees and shrubs generate temporally shaded areas, reducing the solar irradiance input for those plants that grow in the shade and their potential biomass accumulation. Depending on the definition used [1], the area of these transition zones from forested to non-forested matrices, makes up one fifth of the global forested area [2]. Shading of the adjacent non-forested areas in the transition zones, has various effects [1,3]; the reduction of solar irradiance has direct consequences on temperature, evapotranspiration [4,5], and, as a consequence of reduced plant growth and evapotranspiration, it also has an effect on the soil moisture [6,7]. Spatial gradients of solar irradiance change drastically from the zero line of the forest (the forest edge), towards the more open space of the adjacent cropland areas [8,9]. The spatial extent of the shading is influenced by the height of the trees, whereas the transmittance is influenced by the species composition, foliage density, and the type of foliage [7,10,11]. Moreover, the cardinal direction of the edge and azimuth of the sun play an important role, by determining the relative position of the shading element, with respect to the course of the sun [9].

The reaction of plants to increased solar irradiance follows two main pathways: (1) A higher radiation turn-over on plants and soil surface leads to higher temperatures of the plant tissue, which directly affect all physiological processes in plants, including photosynthesis and respiration. (2) Higher temperatures at the leaf level, increase the water vapor deficit of the surrounding air volume and, thus, increase the water loss through the plant's open stomata. Together with an accelerated photosynthesis and the resulting higher water consumption and nutrient demand, the consequence is a higher transpiration rate and water uptake from soil. A sun-lit plant will, therefore, deplete its soil water reservoir, much quicker than a shaded plant, with an added, elevated evaporation taking place in soil that is exposed to sunshine. As transpiration is limited to the availability of soil moisture, plants close their stomata to reduce transpiration as soon as soil water resources fall below the soil's permanent wilting point. A reduction of photosynthesis and growth, including leaf area and yield formation, is the consequence. A reduced leaf area will also affect the growth and yield formation at later stages, when water supply may eventually have returned to sufficient levels again, just by reducing the potential area for the interception of sunlight. This interplay between the solar irradiance, soil moisture availability, evapotranspiration, and plant growth, is complex (Figure 1). However, a detailed understanding of the balance between a sufficient solar irradiance for reaching full growth potential, and the limitations that keep the plant below its potential, is essential, in agricultural systems. As many of these limiting factors have a landscape dimension, a precise knowledge about how limiting factors emerge at the landscape scale and the ecophysiological functioning of these processes, and how and where forested areas and transitions zones may influence these processes, is critical, to upscale the predictions of biomass production, and yields to a landscape level and beyond.

Most crop growth models do not account for the effects of shading at the interception of trees and crops [12]. Some consider management techniques [13] or the impact of climate change [14]. However, the reactions of crops on the limited light resources, are diverse, for the different crop varieties and have to be considered, to model crop growth in these mixed agro-ecosystems [15]. Clearly, there is a competition for light, as reported for wheat [16,17], maize [18], and soybean [19], but these results are often based on studies in agroforestry systems and do not consider the forest edges.

Almost all process-based simulation models for crop growth and development that are employed for impact assessments of food productivity, are one-dimensional. While soil moisture and solar irradiance are essential inputs to drive their simulations, solar irradiance is usually considered to be only dependent on the time of the year and the latitude, and water availability is typically implemented in a way to reflect the small-scale heterogeneity, for example, per square meter. However, at the landscape scale, solar irradiance is affected by terrain and other features of landscape composition—hills and slopes shade cropped areas [20], as do forests and hedges [15,19]. Shading of

plants is implemented in some crop agro-ecosystem models to simulate the simultaneous growth of different plants, for example, in intercropping systems [21–23]. However, to our knowledge, shading parts of the arable land from the adjacent trees or shrubs, is not implemented in any agro-ecosystem models [12] and, hence, is not considered in large-area yield predictions, except for agro-forestry systems [24,25]. Additionally, some agro-ecosystem models are coupled to the hydrological models to reflect the hydrological and soil-specific heterogeneities, when simulating processes at the watershed level [26–28]. However, to our knowledge, none of them account for the altered soil moisture conditions, due to shading in the transition zone between the agricultural land and the forest.

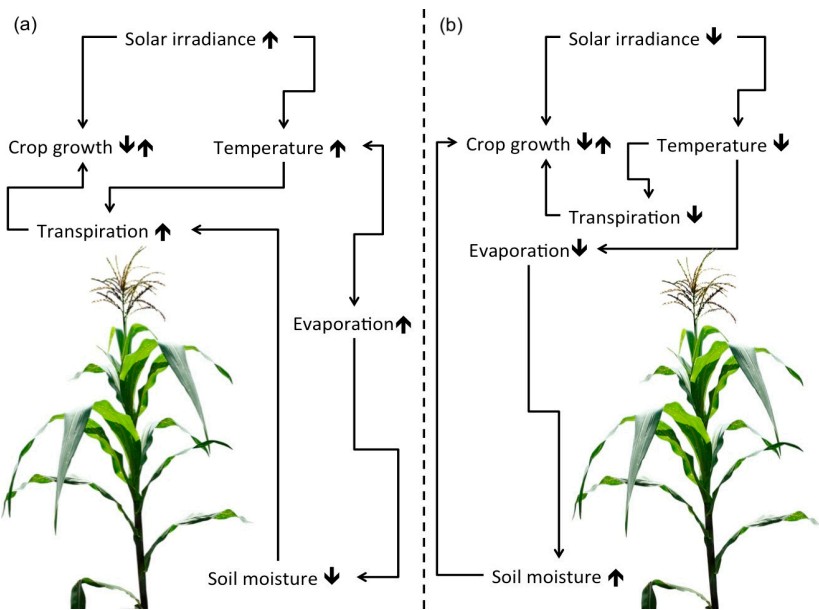

**Figure 1.** Relationship of processes according to higher (**a**) and lower (**b**) solar irradiance that affect the plant growth.

In this study, we simulated the effects of shading by a forest edge on the crop yield and soil moisture of an adjacent field, and investigated the direct effects of reduced irradiance against the indirect effects of the soil moisture. We hypothesized that (i) crop yields within the transition zone is significantly reduced through a reduced solar irradiance, and that (ii) soil moisture feedback plays a significant role in the yield formation in this zone. Generally, yield reduction in transition zones are suggested to be significant in agricultural landscapes. Therefore, we (iii) exemplarily calculated the impact of shading on yield, on the landscape scale for the federal state Brandenburg, Germany, to validate this major hypothesis.

## 2. Methodology

For our approach, we used a virtual landscape model with a forest matrix and surrounding arable land that we simulated in the ArcGIS (ArcGIS Desktop 2011, ArcMAP 10.4.1, ESRI Inc., Redlands, CA, USA). To model solar irradiance, data on the sun angle, geolocation, day of the year, time of the day and height of the shading element were necessary [29]. This simulation included a calculation of the course of the sun for Berlin, Germany (52.500° N, 13.405° E), for one year. The intensity of solar irradiance was corrected to the intensity of the reduced solar irradiance, in the shaded areas. Climate data were combined with the simulated solar irradiance for different distances and cardinal directions from the forest, to simulate the possible yield reductions along the transects, in different directions, from the forest edge into the cropland areas.

### 2.1. Shadow Modeling

#### 2.1.1. Calculation of the Azimuth and the Altitude

To model the course of the sun, daily information of the azimuth and altitude of the sun were necessary. This could be solved with astronomical equations, according to the NOAA (National Oceanic and Atmospheric Administration) Solar Calculator [30]. First, the fractional year (*n*) was calculated:

$$n = \left(\frac{\pi}{180}\right) \times \frac{\left(x - 1 + \left(\frac{\left(\left(\frac{hour + min}{60}\right) - 12\right)}{24}\right)\right)}{365} \tag{1}$$

where *x* is the day of the year. Second, we needed to define the solar declination angle (*declin* as radians) and estimate the equation for time (*timeeq* in min):

$$
\begin{aligned}
declin = \quad & 0.006918 - 0.399912 \times \cos(n) + 0.070257 \times \sin(n) \\
& -0.006758 \times \cos(2n) + 0.000907 \times \sin(2n) - 0.002697 \times \cos(3n) \\
& + 0.00148 \times \sin(3n)
\end{aligned}
\tag{2}
$$

$$timeeq = 229.18 \, (0.000075 + 0.001868 \times \cos(n) - 0.032077 \times \sin(n) - 0.014615 \times \cos(2n) - 0.040849 \times \sin(2n)) \tag{3}$$

Next, we calculated the solar hour angle (*hourangle*) by:

$$hourangle = 15 \times \left( hour + \frac{min}{60} - \frac{(15 - long)}{15} - 12 + \frac{timeeq}{60} \right) \tag{4}$$

where *long* is the longitude (°). Then, we calculated the altitude by:

$$
\begin{aligned}
altitude \ = \sin(x) = \\
\sin(K \times lat) \cdot \sin(K \times declin) \\
+ \cos(K \times lat) \ \times \cos(K \times declin) \ \times \cos(K \times hourangle)
\end{aligned}
\tag{5}
$$

where

$$K = \frac{\pi}{180} \tag{6}$$

and *lat* is the latitude (°). The solar zenith angle (*azimut*) could then be calculated from the hour angle (*hourangle*) and the solar declination (*declin*), using the following equation:

$$azimut \ = \cos(y) = \frac{-(\sin(K \times lat) \times \sin(K \times h) - \sin(K \times declin))}{(cos(K \times lat) \times sin(arccos(sin(K \times h))))} \tag{7}$$

where

$$h = \frac{arcsin(x)}{K} \tag{8}$$

Both, altitude and azimuth were calculated with a two-digit accuracy. For the altitude, the mean absolute variance was 0.22° with a maximum of 0.4°, from 8 am to 4 pm. For the azimuth, the mean absolute variance was 0.14° with a maximum of 0.35°, from 8 am to 4 pm.

#### 2.1.2. Solar Irradiance and the Canopy Transmittance for Modeling

We used the Photovoltaic Geographical Information System (Joint Research Centre Photovoltaic Geographical Information System (PVGIS)) provided by the European Commission, to obtain data on the solar irradiance. We used Berlin, Germany (52.500° N, 13.405° E) as an exemplary location, with an elevation of 35 m.

As the solar irradiance was scattered through the foliage absorption, we used an approximated 0.3 transmittance factor, according to Gholz et al. [10], to simulate the shading from trees. The values

they reported for slash pine (*Pinus elliottii*) were a suitable proxy, as trees in the Berlin area were mostly scots pines (*Pinus sylvestris* L.). This value for reduced solar irradiance was constant for the whole simulation period.

### 2.1.3. Simulation of the Shading Using the Geographical Information Systems

To simulate the shadowing effects with ArcGIS, we used the azimuth, altitude, solar irradiance and reduced solar irradiance, as described above. A block representing a forest with an area of 30 × 30 m and a height of 20 m was implemented into a surrounding flat area of 100 m, in each direction that represented the arable land (Figure 2). The course of the sun was calculated at hourly intervals, for each day of the year, with the ArcGIS (ArcGIS Desktop 2011, ArcMAP 10.4.1, ESRI Inc., Redlands, CA, USA) commands "hillshade" and "shadow". Shadows were simulated pixel-wise (1 × 1 m), writing 1 for the shaded and 0 for the sun-lit areas. In total 4402 separate grid files were calculated and saved. The shadowed areas were multiplied, with an average transmittance factor of 0.3 (see above). Finally, the command "Zonal statistics" was used to generate a query of the hourly reduced solar irradiance, at certain points, around the virtual forest. The points were located in the northern (N), eastern (E), southern (S), and western (W) directions at a distance of 1, 2, 5, 10, 15, 20, 25, 30, 40, 50, 60, 70, 80, 90, and 100 m from the zero line of the forest. The data were stored in a table and were used as the inputs into a crop model.

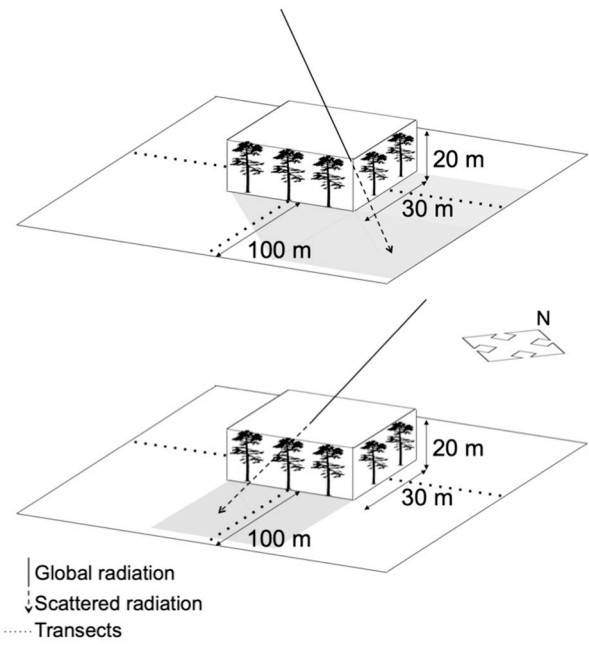

**Figure 2.** Sketch for two exemplary angles of penetrating solar irradiance, through the forest, in a virtual landscape and the shadows it casts. The drawn-through line represents the solar irradiance for Berlin, Germany (52.500° N, 13.405° E). The dashed line denotes the reduced solar irradiance (shadow). The dotted lines represent the directions of the transects for which crop yield was simulated.

### 2.2. Crop Modeling

#### 2.2.1. Simulation

Simulations were conducted using the MONICA (Model for Nitrogen and Carbon in Agro-ecosystems) crop growth simulation model [31]. MONICA works in a one-dimensional way and calculates the values for a minimum spatial resolution of 1 m$^2$ surface area and 2 m depth. On daily time-steps, the crop growth and yield of the silage maize (*Zea mays* L.) and the winter wheat (*Triticum aestivum* L.) was simulated, using thirty years of climate data (see below), each. Every single year was simulated,

independently, without any carry-over or crop rotation effects. This led to $n = 29$ simulations for the winter wheat, with sowing in the first year (1988) of the climate data. For maize, there were $n = 30$ simulations.

Simulations were performed for the plots at different distances (1, 2, 5, 10, 15, 20, 25, 30, 40, 50, 60, 70, 80, 90, and 100 m) from the forest edge, along the cardinal directions (north, east, south, west). This resulted in a total of 3540 observations.

For sowing and harvesting, the implemented functions were used, with an earliest date of sowing for maize, between 4th and 20th of May, and a latest harvest date of 30th of September. For the winter wheat, the sowing period was set from the 5th of October to 8th of November. For both crops, the sowing date of the respective year depended on a minimum temperature of several subsequent days in a row. The sowing dates were based on the farmers' knowledge, and were typical for this region. The parameterization of the MONICA model for wheat and maize was based on several previous studies [32–34].

For simulations of the soil moisture, an initial value of 100% of the field capacity was set. Evapotranspiration was affected by the shading effects, in terms of the reduced solar irradiance, during the simulations. MONICA separated the soil into 10 cm layers. For our simulations we used soil characteristics, as described below, defining three soil horizons (Table 1). The maximum rooting depth for the winter wheat was set to 150 cm, and for the silage maize to 200 cm.

The site was simulated, using the example data implemented in the MONICA. This comprised a latitude of 52.8093° N, no slope, and 0 m above the sea-level. No virtual fertilizers were added and the response to nitrogen was switched off, to avoid it to be limiting. The soil profile had the following characteristics, which were representative of the soils in the region from which the climate data were retrieved—the Federal state of Brandenburg.

**Table 1.** Soil characteristics implemented in the MONICA (Model for Nitrogen and Carbon in Agro-ecosystems). The texture class is given according to the World Reference Base (WRB) for Soil Resources.

| Depth (cm) | Soil Organic Carbon (%) | Texture Class | Raw Density (kg m$^{-3}$) |
|---|---|---|---|
| 0 to 30 | 0.8 | loamy sand | 1446 |
| 30 to 40 | 0.15 | loamy sand | 1446 |
| 40 to 200 | 0.05 | loamy sand | 1446 |

### 2.2.2. Climate Data

Climate data was retrieved from the German Weather Service (DWD) for Berlin-Tegel (station number 430) for the years 1988 to 2017 (30 years). We used average daily wind speed, air temperature, humidity, precipitation as well as the minimum and maximum air temperature. This fulfilled the minimum requirements for the MONICA model. Solar irradiance was added to the climate data, according to our approach, as described above.

To disentangle the effects of soil moisture and solar irradiance, we simulated the yields for the driest and the moistest years. We compared the total annual precipitation for the three driest (1999, 2003, and 2016) and moistest (1993, 2007, and 2017) years, in the respective time periods.

### 2.2.3. Spatial Extent of the Transition Zones

For the calculation of the transition zones in Brandenburg, we used the German Biotope Map. According to the code of land use, the area of transition zones of forest and non-forest land use were calculated using the buffer tool ArcGIS (ArcGIS Desktop 2011, ArcMAP 10.4.1, ESRI Inc., Redlands, CA, USA). The buffered areas were created with different distances, perpendicular to the zero line (20, 50, and 100 m), in both directions, into the interior of either the forest or the non-forest land, and to the outside. After that, these areas were overlaid and clipped. Finally, the share of the transition zones,

compared to the whole area of the respective land-use type (forest: 11,045 km$^2$, non-forest: 18,799 km$^2$), was calculated.

## 3. Results

### 3.1. Solar Irradiance

Solar irradiance increased with increasing distance to the zero line (Figure 3, Table S1). It was least for the north-facing transition zone. Compared to the south-facing transition zone (12.1 MJ m$^{-2}$ day$^{-1}$), which can be considered as fully irradiated; only one-third (4.3 MJ m$^{-2}$ day$^{-1}$) of the irradiance is potentially available close to the zero line in the north. At 30 m, the difference between the north and south, compared to the zero line is only 8%, and almost vanishes from 50 m on.

West- and east-facing transition zones are nearly equal in their average daily solar irradiance. Close to the zero line it is, approximately, two-third (8.5 to 8.6 MJ m$^{-2}$ day$^{-1}$) of the solar irradiance in the south-facing transition zone. At 30 m, the difference to the south-facing transition zone is only 5 to 6% and almost vanishes from 50 m on. An irradiance gradient, with respect to distance to the zero of shading in the south-facing transition zone, is hardly distinguishable.

### 3.2. Soil Moisture

The simulated average soil moisture was highest for both crops, up to ca. 15 m from the zero line, at the north-facing side (Figure 4). Yet, the highest difference between the cardinal directions is at 1 m from the zero line, with ca. 0.19 m$^3$ m$^{-3}$ (north), 0.18 m$^3$ m$^{-3}$ (west and east), and 0.17 m$^3$ m$^{-3}$ (south) for wheat (Figure 4). For maize, the highest difference in soil moisture values were ca. 0.202 m$^3$ m$^{-3}$ (north), 0.199 m$^3$ m$^{-3}$ (west and east), and 0.196 m$^3$ m$^{-3}$ (south). After not more than 50 m from the zero line, soil moisture values did not differ between the shaded and the non-shaded areas, for both crops, with ca. 0.17 m$^3$ m$^{-3}$, for wheat and 0.195 m$^3$ m$^{-3}$ for maize.

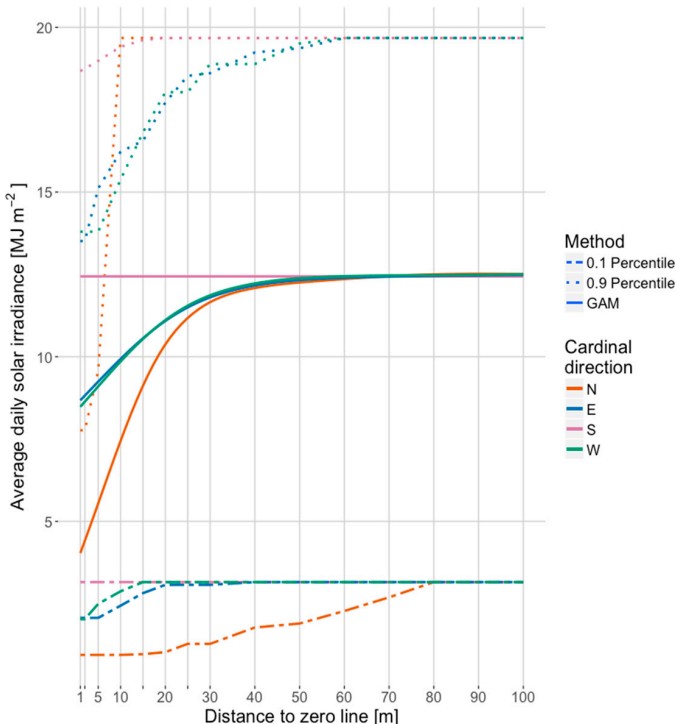

**Figure 3.** Simulated average daily solar irradiance for the four cardinal directions (north, east, south, and west) of the arable land, shaded by a forest, with respect to distance to the zero line. The solid lines represent the general additive models (GAM), fitted to the data, and the dashed lines represent the 0.1 and 0.9 percentile.

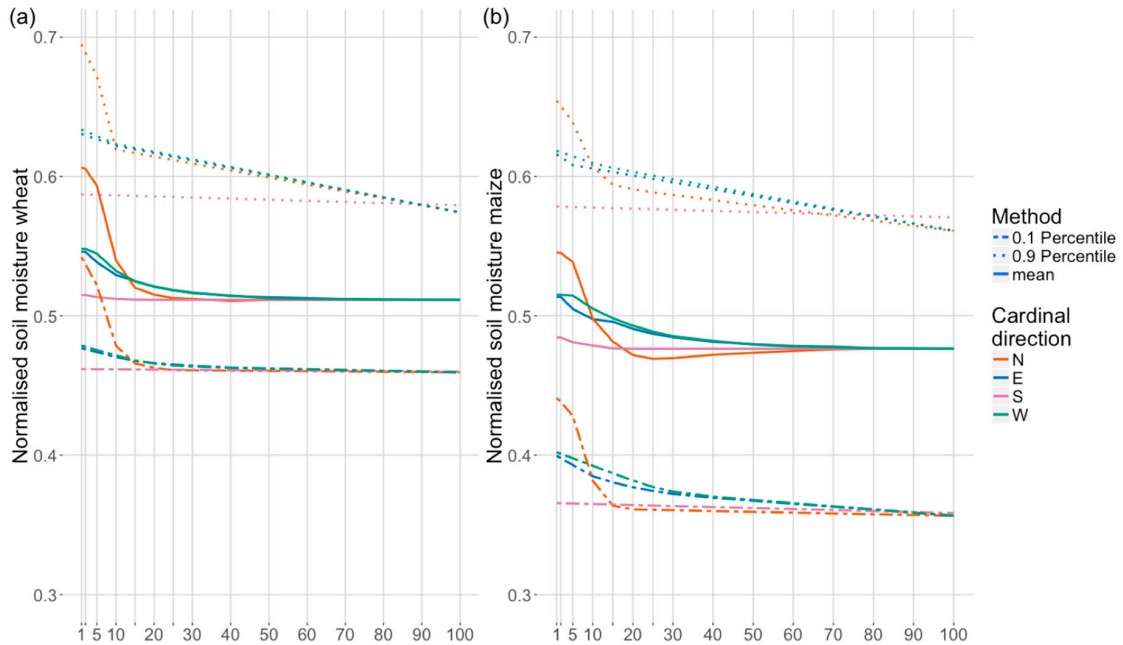

**Figure 4.** Simulated average daily soil moisture for wheat (**a**) and maize (**b**), with respect to the distance to the zero line, and the cardinal direction of a virtual forest matrix shading the adjacent arable land. The solid line represents the mean and the dashed lines represent the 0.1 and 0.9 percentiles.

*3.3. Yield*

Winter wheat yield increased with an increasing distance, up to 10 m from the zero line, in all cardinal directions (Figure 5, Table S2). Yield was 2% lower at 1 m, compared to 10 m, at the south-facing side and 8% lower at the east-facing side. At the west-facing side, the lowest yields were simulated at 1 m, with a reduction of 8%, compared to 15 m. The highest yield reduction of −37%, was simulated for the north-facing side, at 1 m compared to 15 m.

Maize yield increased with the increasing distance from the zero line, in all cardinal directions (Figure 5, Table S2). In the south-facing transition zone, the yield depression was least, with a reduction of 9% in dry mass, at 1 m compared to 15 m. Beyond that, there was nearly no difference in the average yield per year. The west- and the east-facing side had very similar results, where yields were lowest at 1 m. This was a reduction by one-third, compared to the values at 50 m, where the yield was not affected. Thus, a substantial yield reduction occurred at a 1 to 30 m distance (−31% to −4%, compared to 50 m). The highest reduction in yield (−54%), occurred at the north side, between 30 m and 1 m.

Separating the driest and the wettest years from the simulated 30-years time-series, we found that in the three driest years, the average wheat yield was 33% lower, at 1 m in the south, compared to the three wettest years (Figure 6, Table S2). In the east- and the west-facing directions, the difference at 1 m was −11% to −12%. In the north, the yield was 4% lower. In the three wettest years, the yields were 2% lower in the north, up to 10 m from the zero line. At around 50 m, there was no difference between the cardinal directions. We also found that along the north-facing transect, the yields peaked at a 10 m distance, in the dry years, producing 19% more than at the 50 m distance, where the effect ceased. Along the other transects, the yields were only slightly higher and the peak was closer to the zero line. In the wet years, the yield increased towards the interior of the field, along all transects.

At 1 m, the maize yields were 7% higher in the south, in the driest years, compared to the wettest years (Figure 6, Table S2). From 1 m to 10 m, the yields for the driest and the wettest years, were similar, at 10 m. At a greater distance, the yields in the driest years were up to 13% higher, compared to the wettest years in the north and south. From 25 m onwards, the north- and south-facing side reached the yield potential, in the driest years. In the east and the west transects, the yields in the driest years were approximately 4% (15 m) to 7% (70 m) higher, than in the wettest years. From 60 m onwards,

the yields in the wettest years, and from 80 m onwards, the yields in the driest years, were all similar, with a difference of 13%, between treatments. For maize, we observed no yield peaks in the dry years, in the transition zone.

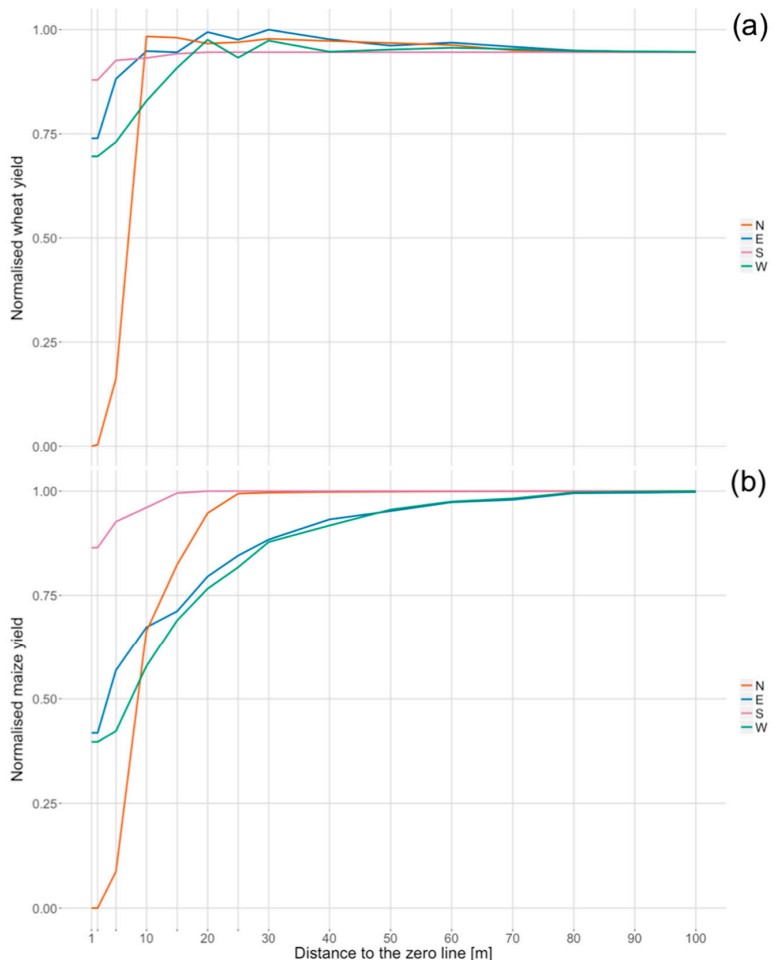

**Figure 5.** Simulated yields for wheat (**a**) and maize (**b**), in the four cardinal directions (north, east, south, and west), with respect to the distance to the zero line and the resulting shading of a forest matrix. The mean values of the annual simulations for 30 years of climate data are shown.

*3.4. Upscaling the Shading Effects on the Crop Yield*

The share of the non-forested area at short distances from the adjacent forests in Brandenburg, is presented in Table 2. Not all non-forested area, but 13,230 km$^2$, is agricultural land, according to the farmers' association (Landesbauernverband Brandenburg); 75% of this area is cropped land, which is about 9923 km$^2$. Maize is grown on 18% (1786 km$^2$) of the cropped land and wheat on 13% of it (1290 km$^2$).

**Table 2.** Calculated share of the transition zone (TZ), in the area of Brandenburg, for the different TZ lengths.

|  | 10 m * | 15 m | 20 m | 30 m * | 40 m * | 50 m | 100 m |
|---|---|---|---|---|---|---|---|
| Non-forest | 4.8% | 6% | 7% | 9.6% | 12% | 15.2% | 26.3% |

Values with asterisks (*) were interpolated linearly between the modeled values of those without asterisks.

Considering the area of the crops in the Brandenburg, the simulated values for the yield from the MONICA model, the length and share of the transition zone and an, approximated yield reduction in

the transition zone, we arrived at an overall yield loss of 5.4% for wheat and 8.5% for maize, due to the shading by the forest edges (Table 3).

**Table 3.** Calculation of the yield reduction due to shading of the forest edges, for the winter wheat (grain yield) and the silage maize (whole plant), in Brandenburg. TZ shares were taken from Table 2, TZ length and yield reduction in TZ are the calculated values in this article. Yield reduction in Brandenburg was calculated, in comparison to no-shading in TZ.

| Crop | Area (ha) | Average Yield (kg ha$^{-1}$) | TZ (m) | TZ Share (%) | Yield Reduction in TZ (%) | Yield Reduction in Brandenburg (%) |
|---|---|---|---|---|---|---|
| Winter wheat | 129,000 | 6314 | 15 | 6 | 10.4 | 5.4 |
| Silage maize | 178,600 | 2564 | 30 | 9.6 | 12.3 | 8.4 |

TZ = transition zone.

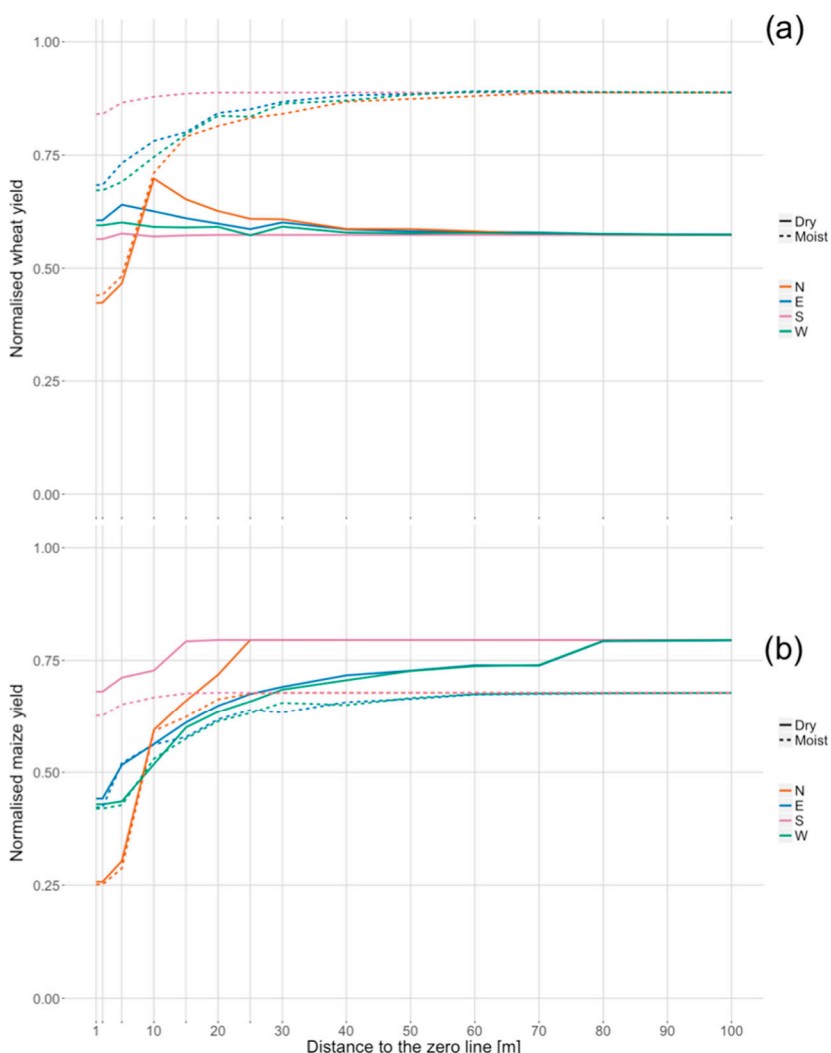

**Figure 6.** Simulated yields for wheat (**a**) and maize (**b**) in the four cardinal directions (north, east, south, and west), with respect to distance to the zero line and the resulting shading of a forest matrix. The solid line represents the mean for the three driest years and the dashed line represents the mean of the three wettest years, within the 30 years of climate data.

## 4. Discussion

### 4.1. Relation between Yield Reduction, Soil Moisture, and Solar Irradiance

In general, the gradients of yield reduction (Figure 5) correlated with the modeled solar irradiance (Figure 3). The reduction of solar irradiance was the strongest in the north, and the spatial extent of shading was similar in the east and west. The general shapes of the gradients for the simulated yields, as well as solar irradiance, were similar and the solar irradiance could be adduced as a major explanatory variable for the decreasing yields, in the transition zone. Although a gradient was also simulated for the soil moisture, it was very small.

To disentangle the effects of soil moisture and solar irradiance, we separated the three driest and three moistest years, from the simulations, under the assumption that in the wet years, crop growth would not be limited by water deficiency, but only by radiation. Indeed, we found that yields for wheat were reduced in the shaded transition zone, while outside, towards the field, simulated wheat yields remained stable. In the dry years, as a contrast, wheat was limited by water availability, along the transect, expressed by a lower yield level, in the simulations. However, in the northern transition zone, we observed higher simulated yields, at 5–40 m distance, from the zero line, as compared to the field matrix. A similar effect, but a less pronounced one was also observed for the western and the eastern transect.

Maize, obviously, grew better in drier than in the wetter years in the simulations, as the yields were higher in the drier years compared to the moistest years, which reflects the nature of maize, which is better adapted to warm and dry conditions than wheat [5]. In the shaded transition zone, yields were equal to the moistest years, indicating that the yields, at no time, were limited by water, but were significantly limited by a reduced radiation input in the transition zone.

With a comparably simple set-up of tools and a minimum of climate data, it was already possible to model the effect of the forested areas on solar irradiance, soil moisture, and crop yields, in the transition zones, and the feedback loop that coupled soil moisture and crop physiology. However, there were further effects in the transition zones that we were not able to model, including wind turbulences, cardinal direction of wind shelter, rain shadow, or snow entrapment [5,7]. As this shelter effect was reported to have a positive influence on the soil moisture content [5], future work should obtain data on, for example, the transitional gradients of wind speed. In addition, the yields in the transition zones might have been affected by allelopathy and nutrient deficiencies [7], which was not modeled as well. In a natural environment, these effects might have compensated the yield reductions through shading.

The transmittance of solar irradiance through the trees in the transition zone of forests, was approximated with 30% for pine [10], in our approach. However, Voicu and Comeau [11] reported a 60% transmittance for aspen, close to the zero line and at least 80% after 1 H—an often used concept of tree heights—where 1 H equals the height of the element that shadows. Hence, the tree species composition should be considered when solar irradiance is modeled.

### 4.2. Magnitude of the Shading Effects on the Yield

Considering the whole transition zone in all cardinal directions, the simulated yield reduction in transition zones was, approximately, 10% for wheat and 12% for maize. Simulated yields were lower, closer to the forest. In all cardinal directions, the yields were reduced by a reduced solar irradiance, compared to the areas with full irradiance. The yield reduction was highest in the north and lowest in the south. The simulated results were in line with measurements. Mitchell et al. [35] measured very similar gradients for soybean, with respect to the distance to the zero line, as the ones simulated close to the forest with a decrease of ca. 30% to 50% within 100 m. The magnitude of the simulated yield reduction for wheat was similar to the 34% reported by Malik and Sharma [36], and the 31% reported by Lyles et al. [37]. Dufour et al. [12] reported a decrease of 50%, due to a reduced number of grains per spike and weight of grains. Nuberg [5] measured similar, and even higher, yield reductions for wheat.

*4.3. The Spatial Extent of the Shading Effects on the Yield*

Most of the yield reduction for the simulated wheat occurred within 15 m. The impact of shading on the maize yields occurred at least until 30 m from the zero line. The simulated solar irradiance reached maximum values at, approximately, 30 m. Correspondingly, the effects on soil moisture occurred until a distance of 30 m. Thus, our model predicted a correlation between the solar irradiance, soil moisture, and a reduced yield, to a distance of 15 m from the zero line, on average, and 30 m maximum, depending on the cardinal direction. These distances were in line with findings from other authors. They often use the concepts of tree heights H, which were as follows, in our approach: 15 m = 0.75 H and 30 m = 1.5 H. Voicu and Comeau [11] reported a maximum of 1.5 H (aspen stand) for reduced solar irradiance, while Emmingham and Waring [38] reported 1 H (Douglas-fir). Lyles et al. [37] reported a decrease in the winter wheat yield, between 0.25 and 3 H. In a review by Kort [7], values from 1 to 3.3 H were reported by several authors and for different crops. Bulir [39]) even reported a yield reduction up to a distance of 300 m (20 H), for wheat. Our results did not account for distances that went beyond 100 m.

The distances from the forest edge for reduced solar irradiance, soil moisture, and reduced yields depended on the cardinal direction. The lowest impact, with respect to distance, on solar irradiance and yield, was in the south of the forest and the shrub matrices, for the sites on the northern hemisphere of the globe. In the north, it was highest, while it was intermediate in the east and the west. This effect was also found by Matlack [9], for soil moisture and solar irradiance, by Voicu and Comeau [11], as well as by Groot and Carlson [40]. With 30 m, the transition zone for our case study of modeled solar irradiance, was only one-third of the often referenced value of 100 m for the forested transition zones [2,41,42]. This suggests that this value is not universally applicable, especially, not for agricultural areas.

Accepting the approximation we made to simulate the shading effects of the forest edges on wheat and maize, our results showed a regional yield loss of 5% to 8%. These calculations helped to understand the importance of transition zones in agricultural landscapes. Although it is a simplified calculation, it combined the magnitude of yield loss, with the spatial extent for a whole region. However, these calculations did not account for the biological effects that are beneficial for plant growth, like pollination or pest control. The benefits and yield gains that were in relation to the distance to forests were likely to be much higher, compared to the loss through shading. Several aspects, like a higher biodiversity or the higher rate of ground water recharge of and through forests—just to mention some—were likely to have a higher importance, in terms of provision of the ecosystem services. Although the provision of food was one of the most important ecosystem services, a higher biodiversity in the transition zones, likely compensated the yield loss through shadows. With that in mind, management implications for farmers might be derived from the results, for the selection of crop varieties planted adjacent to the forests. Moreover, knowledge about the yield reduction might encourage farmers to reduce the management intensity adjacent to forests in favor of other ecosystem services rather than food provision.

## 5. Conclusions and Outlook

We used a model to assess the yield reductions due to shading in the transition zones of agricultural landscapes, and used the simulation model to investigate the feedback loops that apply at the different distances from the zero line. With a minimum set of climate data, modeled solar irradiance, the crop growth model MONICA, and the basic functions of a geographical information system, the transitional gradients were modeled. In this paper, we presented and explained the procedure, analyzed the results, and compared them with the measured results from the literature. Moreover, we estimated the spatial extent of the transition zones in an agricultural landscape—Brandenburg in Germany—and calculated the yield loss on a regional level. We found the following, as stated below:

- Solar irradiance and yield have a strong correlation; with increasing distance to forest, solar irradiance and yield increase.
- The main influencing factors for the reduction of solar irradiance and the accompanying yield are tree height, distance to the forest, and cardinal direction.
- Crop varieties react differently, according to their physiological disposition.
- In dry years, the shading effects in the transition zones can be beneficial for the crop growth.
- On a regional level, a yield reduction of 5% to 8% can be considered to have been caused by shading in the transition zones.

Although our approach is satisfying to account for yield reductions due to shading in agricultural landscapes, it could also be further developed. The transmittance rates could be included as a logistic regression function of the distance to the forest, to make the results more precise. Moreover, functions for the transitional gradients of temperatures, wind speed, and soil moisture could be implemented to allow more precise simulations, on a larger scale. Further, a step from virtual to real landscapes would be to use light detection and ranging (LIDAR)-based tree maps, to simulate shadows in the agricultural landscapes. This would make the regional prediction of yield loss, more precise, especially, according to the cardinal directions. With respect to the ecosystem services approach, a comparison of trade-offs and benefits between the yield losses and, for example, biodiversity would help to understand the impact of the forest transition zones and to improve the management decisions.

**Supplementary Materials:** The following are available online at http://www.mdpi.com/2077-0472/9/1/6/s1, Table S1: Simulated solar irradiance (MJ m$^{-2}$) for the four cardinal directions (north (N), east (E), south (S), and west (W)) with respect to distance (m) to the zero line (Dist). The values represent the yearly average, Table S2: Simulated maize and wheat yields (kg ha$^{-1}$), as well as soil moisture (m$^3$ m$^{-3}$) for the four cardinal directions (*Dir*; north (N), east (E), south (S), and west (W)), with respect to the distance (m) to the zero line (*Dist*). The values represent an average for 30 years (Wheat *n* = 29, Maize *n* = 30). Dry and Moist are the means of the three driest and moistest years, in the climate time-series.

**Author Contributions:** Conceptualization, M.S., C.N. and G.L.; Methodology, M.S., C.N. and R.F.; Software, M.S., C.N. and R.F.; Validation, C.N., M.G.F.M. and R.F.; Formal Analysis, M.S.; Investigation, M.S.; Data Curation, M.S.; Writing—Original Draft Preparation, M.S. and C.N.; Writing—Review and Editing, M.S., C.N., M.G.F.M., R.F. and G.L.; Visualization, M.S.; Supervision, C.N., M.G.F.M. and G.L.

**Acknowledgments:** We are very grateful to several people who helped to collect and analyze the data and who provided valuable suggestions concerning the study design and the virtual landscape: Tommaso Stella, Lidia Völker, Carola Voigt, Tomas Selecky, Evelyn Wallor, Dennis Melzer, Kurt-Christian Kersebaum, and Hubert Jochheim. We would like to thank the anonymous reviewers for their very constructive and helpful comments.

**Conflicts of Interest:** The authors declare no conflict of interest.

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
