# Peer review of "Modeling Yields Response to Shading in the Field-to-Forest Transition Zones in Heterogeneous Landscapes"

_agriculture, doi:10.3390/agriculture9010006_

Reviewer 1 Report

The authors present an interesting research for crop yield modelling in transition zones in heterogeneous landscapces. Overall, the investigation is properly approached and conducted, whilst the document is well structured and written. Still, there are some concerns that should be addressed to improve this work:

1) Although the novelty of the paper is mentioned several times, the introduction section lacks a explicit paragraph devoted to critically review several previous and similar papers, in order to clearly highlight how this study contributes to make progress in the SoTA in which it is framed.

2) The quality and resolution of the data used in ArcGIS are not provided. For instance, the "hillshade" tool requires a raster (DEM) as an input; however, no information about this is provided.

3) The visual impact of the results might be enhanced by adding some of the maps produced using ArcGIS.

4) The terminology used in the discussion section is wrong. There is no statistical evidence about the relationships between parameters, so that authors cannot claim that some of them are correlated to each other. For that purpose, the authors should implement some dependence measures depending on the nature of the data (Pearson, Spearman, Phi, etc.).

Author Response

1) Although the novelty of the paper is mentioned several times, the introduction section lacks a explicit paragraph devoted to critically review several previous and similar papers, in order to clearly highlight how this study contributes to make progress in the SoTA in which it is framed.

Thank you very much for this comment. In the paragraph starting in line 87 we reviewed recent literature on process-based crop models to compare those models with the implementations we want to make for the MONICA model regarding shadows of landscape elements. We have added now a smaller body of literature references that also frame the empricial evidence of shading effects and some similar modelling approaches from other agricultural systems.

2) The quality and resolution of the data used in ArcGIS are not provided. For instance, the "hillshade" tool requires a raster (DEM) as an input; however, no information about this is provided.

Within section 2.1.3 we describe the digital elevation map having 1x1 m pixels, a block of 20 m height that represents the digital elevation and a surrounding area of 100 m in each cardinal direction. This is in total 230 m. There was no real map as input, as we simulated a fully virtual landscape, except for the input of solar radiation data. 

3) The visual impact of the results might be enhanced by adding some of the maps produced using ArcGIS.

During this process, no maps were produced. To enhance the visual impression, we have exemplarily added two shadow casts in Figure 2.

4) The terminology used in the discussion section is wrong. There is no statistical evidence about the relationships between parameters, so that authors cannot claim that some of them are correlated to each other. For that purpose, the authors should implement some dependence measures depending on the nature of the data (Pearson, Spearman, Phi, etc.).

Admittedly, the wording was misleading in the discussion. We edited this section a bit and deleted the word 'correlation'. To our point of view, a visual description of graphs is sufficient for our argumentation.

Reviewer 2 Report

Summary:

The aim of the study was to simulate the effects of shading by a forest edge on crop yield and soil moisture of an adjacent field and investigate the direct effects of reduced irradiance against the indirect effect of the soil moisture. It was hypothesised that the crop yields within the transition zone is significantly reduced through reduced solar irradiance and that the soil moisture feedback plays a significant role in the yield formation in this zone. Thus, the impact of shading on yield on the landscape scale was calculated. It was concluded that the solar irradiance and yield have a strong correlation; with increasing distance to forest solar irradiance and yield increase. The main influencing factors for the reduction of solar irradiance and the accompanying yield are tree height, distance to the forest and cardinal direction. Crop varieties react differently according to their physiological disposition. In dry years, the shading effects in transition zones can be beneficial for crop growth. On a regional level, a yield reduction of 5 to 8 % can be considered caused by shading in transition zones. Beside the very good concept some specific comments can be raised.

Specific comments:

1. All of the equations should be numbered properly which will be beneficial in their further referring. 

2. In line 188 authors do refer to set effective rooting depth for maize and wheat. Are you sure about this values? Are they not too optimistic? More importantly, any of the values used for modeling needs to be justified? Reference, recommendations and so on. This should be applied at all input values especially in case of modeling.

3. Figure 4. wheat and maize should be in brackets or any other style of division from text whould be beneficial.

4. Table 3. Average yields for winter wheat and silage maize. It is in grains or it is biological yield of biomass? Thus, are you sure about those numbers?

5. As it was depicted above, mostly justification and clarification of input parameters for MONICA crop growth simulation should be specified.

Final judgement:

The study comprehensively describe the improvement into currently developed modeling system by incorporation of landscape shading of crops. Therefore, novelty of the approach is undeniable. However, some of the aspects needs to be described / clarified in closer details which would help to prevent any potential inaccuracies of developed motel utilisation. In this terms, minor revision can be recommended.

Author Response

1. All of the equations should be numbered properly which will be beneficial in their further referring. 

Done.

2. In line 188 authors do refer to set effective rooting depth for maize and wheat. Are you sure about this values? Are they not too optimistic? More importantly, any of the values used for modeling needs to be justified? Reference, recommendations and so on. This should be applied at all input values especially in case of modeling.

Thank you very much for this comment. There was a mistake in the nomenclature. The input values for the rooting depth are maximum values and not the effective values. They might be too optimistic. However, the root growth in MONICA depends mainly on the soil temperature, the soil texture and a root growth model. Our approach was to not have too many factors that limit the root growth. That's why we used these rather optimistic values.

For other values we specified the source. 

3. Figure 4. wheat and maize should be in brackets or any other style of division from text whould be beneficial.

Done.

4. Table 3. Average yields for winter wheat and silage maize. It is in grains or it is biological yield of biomass? Thus, are you sure about those numbers?

Thanks for the hint. Wheat is measured in grain yield and silage maize is harvested as whole plant. We added this to the description of the table. We are sure about these values as we took them from the federal statistical office and from the farmers association, but these sources are in German only.

5. As it was depicted above, mostly justification and clarification of input parameters for MONICA crop growth simulation should be specified.

We have added a bit more background on the MONICA model and its evidence-based parameterisation for wheat and maize crops in Section 2.2.1.